# Optimal Filtering of Markov Jump Processes Given Observations with State-Dependent Noises: Exact Solution and Stable Numerical Schemes

**Andrey Borisov [1],\* and Igor Sokolov [2]**

[1]  Institute of Informatics Problems of Federal Research Center "Computer Science and Control" RAS, 44/2 Vavilova str., 119333 Moscow, Russia

[2]  Faculty of Computational Mathematics and Cybernetics, Lomonosov Moscow State University, GSP-1, 1-52 Leninskiye Gory, 119991 Moscow, Russia; ISokolov@cs.msu.ru

\*  Correspondence: ABorisov@frccsc.ru

**Abstract:** The paper is devoted to the optimal state filtering of the finite-state Markov jump processes, given indirect continuous-time observations corrupted by Wiener noise. The crucial feature is that the observation noise intensity is a function of the estimated state, which breaks forthright filtering approaches based on the passage to the innovation process and Girsanov's measure change. We propose an equivalent observation transform, which allows usage of the classical nonlinear filtering framework. We obtain the optimal estimate as a solution to the discrete–continuous stochastic differential system with both continuous and counting processes on the right-hand side. For effective computer realization, we present a new class of numerical algorithms based on the exact solution to the optimal filtering given the time-discretized observation. The proposed estimate approximations are stable, i.e., have non-negative components and satisfy the normalization condition. We prove the assertions characterizing the approximation accuracy depending on the observation system parameters, time discretization step, the maximal number of allowed state transitions, and the applied scheme of numerical integration.

**Keywords:** stochastic differential observation system; nonlinear filtering problem; state-dependent observation noise; numerical filtering algorithm; filtering given time-discretized observations; stable approximation; approximation accuracy

---

## 1. Introduction

The Wonham filter [1], as well as the Kalman–Bucy filter [2], is one of the most practically used filtering algorithms for the states of the stochastic differential observation systems. It is applied extensively for signal processing in technics, communications, finance and economy, biology, medicine, etc. [3–6]. The filter provides the optimal in *the Mean Square* (MS) sense on-line estimate of the finite-state *Markov Jump Process*. (MJP) given indirect continuous-time observations, corrupted by the Wiener noise. The elegant algorithm represents the desired estimate as a solution to *a Stochastic Differential System* (SDS) with continuous random processes on *the Right-Hand Side* (RHS).

The fundamental condition for the solution to the filtering problem is the independence of the observation noise intensity of the estimated state. It provides the continuity from the right for the natural flow of $\sigma$-algebras induced by the observations, with subsequent utilization of the innovation process framework. The condition violation breaks these advantages. In the case of the state-dependent observation noise, the author of [7] presents the optimal estimate within the class of the linear estimates. Further, the authors of [8,9] use filters of a linear structure for the solution to the $\mathcal{H}_2$-optimal state filtering problem. To find the absolute optimal filtering estimate, one has to

make extra efforts. First, for proper utilization of the stochastic analysis framework, one needs to reformulate the optimal filtering problem, "smoothing forward" the flow of $\sigma$-algebras induced by the observations. Second, in the case of state-dependent noise, the innovation process contains less information than the original observations. One has to supplement the innovation by the observation quadratic characteristic, which represents a continuous-time noiseless function of the estimated MJP state. In general, the optimal filtering given partially noiseless observations is a challenging problem. Its solution can be expressed either as a sequence of some regularized estimates [10] or by the additional differentiation of the smooth observation components or their quadratic characteristics [11–14]. In both cases, one needs to realize a limit passage, which is difficult in computers.

Even in the traditional settings, the numerical realization of the MJP state filtering is a complicated problem. For example, the explicit numerical methods based on the Itô–Taylor expansion applied to the Wonham filter equation, diverge: the produced approximations do not meet component-wise non-negativity condition. Over time the approximation components reach arbitrary large absolute values. Further, in the presentation, we refer to the approximations, preserving both the component non-negativity and normalization condition as *the stable ones*.

The Wonham filtering equation is a particular case of the nonlinear Kushner–Stratonovich equation. To solve it, one can use various numerical algorithms

- the procedures based on the weak approximation of the original processes by Markov chains [15,16],
- some variants of the splitting methods [17],
- the robust procedures based on the Clark transform [18,19],
- the schemes, which represent the conditional probability distributions through the logarithm [20], etc.

All the algorithms are developed for the case of additive observation noise and based on the Girsanov's measure transform. Hence, they are useless for the estimation of the MJP given the observations with state-dependent noise.

The goal of the paper is two-fold. First, it presents a theoretical solution to the MS-optimal filtering problem, given the observations with state-dependent noise. Second, it introduces a new class of stable numerical algorithms for filter realization and investigates its accuracy. We organize the paper as follows. Section 2 contains a description of the studying observation system with state-dependent observation noise along with the MS-optimal filtering problem statement. To solve the problem, one needs to transform the available observations both to preserve the information equivalence and suit for application of the known results of the optimal nonlinear filtering. Section 3 describes both the observation transformation and the SDS defining the optimal filtering estimate. The SDS is discrete–continuous and contains both continuous and counting random processes on the RHS. Previously, the author of the note [21] presents a sketch of the observation transform, but it cannot guarantee the uniqueness of that SDS solution.

Section 4 presents a new class of the stable numerical algorithms of the nonlinear filtering. The main idea is to discretize original continuous-time observations and then find the MS-optimal filtering estimate given the sampled observations. The authors of [22] use this idea to solve a particular case of the estimation problem, namely the classification problem of a finite-state random vector given continuous-time observations with multiplicative noise. Section 4.1 contains a general solution to the problem. The corresponding estimate represents a ratio, which numerator and denominator are the infinite sums of integrals. They are shift-scale mixtures of the Gaussians. The mixing distributions, in turn, describe the occupation time of the system state in each admissible value during the time discretization interval. In Section 4.2, we suggest approximating the estimates by a convergent sequence bounding number $s$ of possible state transitions, which occurred over the discretization interval. We replace the infinite sums in the formula of the optimal estimate by their finite analogs and also investigate the accuracy of the approximations. We refer these approximations as *the analytical ones of the s-th order*. One cannot calculate the integrals analytically and have to replace them with some

integral sums, and this brings an extra error. Section 4.3 analyzes the value of this error and the total distance between the optimal filtering estimate given the discretized observations and its numerical realization. Section 4.4 presents a numerical example that illustrates the conformity of theoretical estimates and their numerical realization. Section 5 contains discussion and concluding remarks.

## 2. Continuous-Time Filtering Problem Statement

On the probability triplet with filtration $(\Omega, \mathcal{F}, \mathcal{P}, \{\mathcal{F}\}_{t \geqslant 0})$ we consider the observation system

$$X_t = X_0 + \int_0^t \Lambda^\top(s) X_s ds + M_t^X, \tag{1}$$

$$Y_t = \int_0^t f(s) X_s ds + \int_0^t \sum_{n=1}^N X_s^n G_n^{1/2}(s) dW_s. \tag{2}$$

Here

- $X_t = \mathrm{col}(X_t^1, \ldots, X_t^N) \in \mathbb{S}^N$ is an unobservable state which is a finite-state *Markov jump process* (MJP) with the state space $\mathbb{S}^N \triangleq \{e_1, \ldots, e_N\}$ ($\mathbb{S}^N$ stands for the set of all unit coordinate vectors of the Euclidean space $\mathbb{R}^N$) with the transition matrix $\Lambda(t)$ and the initial distribution $\pi = \mathrm{col}(\pi^1, \ldots, \pi^N)$; the process $M_t^X$ is an $\mathcal{F}_t$-adapted martingale,
- $Y_t = \mathrm{col}(Y_t^1, \ldots, Y_t^M) \in \mathbb{R}^M$ is an observation process: $W_t = \mathrm{col}(W_t^1, \ldots, W_t^M) \in \mathbb{R}^M$ is an $\mathcal{F}_t$-adapted standard Wiener process characterizing the observation noise, $f(t)$ is an $M \times N$-dimensional observation matrix and the collection of $M \times M$-dimensional matrices $\{G_n(t)\}_{n=\overline{1,N}}$ defines the conditional observation noise intensities given $X_t = e_n$.

The natural flow of $\sigma$-algebras generated by the observations $Y$ up to the moment $t$ is denoted by $\mathcal{Y}_t \triangleq \sigma\{Y_s : s \in [0,t]\}$, $\mathcal{Y}_0 \triangleq \{\varnothing, \Omega\}$.

*The optimal state filtering given the observations $Y$ is to find the Conditional Mathematical Expectation* (CME)

$$\widehat{X}_t \triangleq \mathbf{E}\{X_t | \mathcal{Y}_{t+}\}. \tag{3}$$

## 3. Observation Transform and Optimal Filtering Equation

Before derivation of the optimal filtering equation we specify the properties of the observation system (1) and (2).

1. All trajectories of $\{X_t\}_{t \geqslant 0}$ are continuous from the left and have finite limits from the right, i.e., are *cádlág-processes*.
2. Nonrandom matrix-valued functions $\Lambda(t)$, $f(t)$ and $\{G_n(t)\}_{n=\overline{1,N}}$ consist of the *cádlág*-components.
3. The noises in $Y$ are uniformly nondegenerate [10], i.e., $\min\limits_{\substack{1 \leqslant n \leqslant N, \\ t \geqslant 0}} G_n(t) > \alpha I$ for some $\alpha > 0$; here and after, $I$ is a unit matrix of appropriate dimensionality.
4. The processes

$$K_{ij}(t) \triangleq \mathbf{I}_{\{0\}}(G_i(t) - G_j(t)), \quad i,j = \overline{1,N} \tag{4}$$

have a finite variation; here and after, $\mathbf{I}_{\mathcal{A}}(x)$ is an indicator function of the set $\mathcal{A}$, and $\mathbf{0}$ is a zero matrix of appropriate dimensionality.

Conditions 1–3 are standard for the filtering problems [10]. They guarantee the proper description of MJP distribution $\pi(t) \triangleq \mathbf{E}\{X_t\}$ by the Kolmogorov system $\pi(t) = \pi + \int_0^t \Lambda^\top(s) \pi(s) ds$. Condition 4 relates to the quadratic characteristic of the observation process as a key information source itself. Below we show that collection of $G_n(\cdot)$, distinguished for different $n$, allows to restore the state $X_t$ *precisely* given the available *noisy* observations. Condition 4 guarantees the local regularity of the time subsets, where $G_n(\cdot)$ coincide and/or differ each other: one can express them as finite unions of

the intervals. The condition is not too restrictive: for instance, they are valid when $G_n(\cdot)$ are piece-wise continuous with bounded derivatives.

Both the system state and observation are special square-integrable semimartingales [6,23] with the predictable characteristics

$$\langle X, X \rangle_t \triangleq X_t X_t^\top - \int_0^t X_{s-} dX_s^\top - \int_0^t dX_s X_{s-}^\top =$$
$$= \int_0^t \left( \mathrm{diag}\left( \Lambda^\top(s) X_s \right) - \Lambda^\top(s)\, \mathrm{diag}\, X_s - \mathrm{diag}\,(X_s)\,\Lambda(s) \right) ds \quad (5)$$

and

$$\langle Y, Y \rangle_t \triangleq Y_t Y_t^\top - \int_0^t Y_{s-} dY_s^\top - \int_0^t dY_s Y_{s-}^\top = \sum_{n=1}^{N} \int_0^t X_s^n G_n(s) ds. \quad (6)$$

Conditions 1–3 and the properties of $X_t$ guarantee $\mathcal{P}$-a.s. fulfilment of the following equalities for the one-sided derivatives of $\langle Y, Y \rangle_t$:

$$\left. \frac{d\langle Y,Y \rangle_s}{ds} \right|_{s=t-} = \sum_{n=1}^{N} X_{t-}^n G_n(t-) = \sum_{n=1}^{N} X_t^n G_n(t-),$$

$$\left. \frac{d\langle Y,Y \rangle_s}{ds} \right|_{s=t+} = \sum_{n=1}^{N} X_{t-}^n \left( G_n(t-) + \Delta G_n(t) \right) = \sum_{n=1}^{N} X_t^n G_n(t), \quad (7)$$

where $\Delta G_n(t) \triangleq G_n(t) - G_n(t-)$ is a jump function of $G_n(t)$. So, if there exists a nonrandom instant $t^* > 0$ such that $\sum_{n=1}^{N} \pi^n(t^*) \Delta G_n(t^*) \neq 0$, then $\mathcal{Y}_{t^*} \subset \mathcal{Y}_{t^*+} = \mathcal{Y}_{t^*} \vee \sigma\{\sum_{n=1}^{N} X_{t^*}^n \Delta G_n(t^*)\}$. The inclusion presumes the flow of $\sigma$-subalgebras $\{\mathcal{Y}_t\}_{t \geqslant 0}$ is not necessarily continuous from the right for the considered observations [24]. This is a reason to define a filtering estimate as a CME of $X_t$ with respect to the "smoothed" flow $\mathcal{Y}_{t+}$ for subsequent correct usage of the stochastic analysis framework.

Let us transform the available observations in such a way to derive the optimal filtering estimate by the standard methods [6,23]. Initially, the idea of this transform is suggested in [11]. As the result, the authors introduce the pair

$$U_t \triangleq \int_0^t \left( \left. \frac{d\langle Y,Y \rangle_u}{du} \right|_{u=s+} \right)^{-1/2} dY_s, \quad (8)$$

$$\langle Y, Y \rangle_t = \sum_{n=1}^{N} \int_0^t X_s^n G_n(s) ds. \quad (9)$$

The authors of [11] prove coincidence of the $\sigma$-algebras $\mathcal{Y}_t = \sigma\{U_s,\ 0 \leqslant s \leqslant t\} \vee \sigma\{\langle Y, Y \rangle_s,\ 0 \leqslant s \leqslant t\}$ for the general diffusion observation systems. However, they do not pay attention to the continuity of $\{\mathcal{Y}_t\}$ from the right. The authors of [12,14] suggest to replace the observations $\langle Y, Y \rangle_t$ by their derivative

$$Q(t) \triangleq \left. \frac{d\langle Y,Y \rangle_s}{ds} \right|_{s=t-} = \sum_{n=1}^{N} X_{t-}^n G_n(t-). \quad (10)$$

Then, one can construct the optimal estimate either to use $Q_t$ as a linear constraint or to differentiate (10) for extraction of the dynamic noises. The papers [12,14] contain a rather pessimistic conclusion: the number of differentiations is unbounded in the general case of diffusion observation system. In contrast, we estimate a finite-state MJP and can construct the optimal filtering estimate using $Q$ without additional differentiation.

So, the transformed observations will contain

- diffusion processes with the unit diffusion,
- counting stochastic processes,
- indirect state observations obtained at the nonrandom discrete moments.

The first transformed observation part is the process $U_t$ (8), and in view of (2) and (7) it can be rewritten as

$$U_t = \int_0^t \overline{f}(s) X_s ds + \overline{W}_t, \tag{11}$$

where $\overline{f}(s) \triangleq \sum_{n=1}^N G_n^{-1/2}(s) f(s) \operatorname{diag}(e_n)$ and $\overline{W}_t$ is an $\mathcal{F}_t$-adapted standard Wiener [10].

The process $Q_t$ could play the role of the second part of the transformed observations since $\mathcal{Y}_t = \sigma\{U_s, Q_s, \ s \in [0,t]\}$ [11], however the natural flow of $\sigma$-algebras generated by the couple $(U, Q)$ is not continuous from the right yet. Moreover, the process $Q_t$ is matrix-valued and looks overabundant for the filter derivation. The point is, $Q_t = Q(t, X_{t-})$ (10) is a function of the finite-set argument $X_t$, and it affects the estimate performance through its complete preimage

$$Q_t = Q(t, X_{t-}) \xrightarrow{Q^{-1}} \{e_n \in \mathbb{S}^N : \ G_n(t-)e_n = Q_t\}.$$

To go to the preimage we introduce the following transformation of $Q_t$:

$$H_t \triangleq \sum_{n=1}^N \mathbf{I}_{\{0\}} (Q_t - G_n(t)) e_n.$$

$H_t$ is a $\mathcal{Y}_t$-adapted vector process with components 0 or 1, but the trajectories $H_t$ are not *cádlág* processes. Due to the fact $X_{t-} = X_t$ $\mathcal{P}$-a.s. for $\forall\, t \geq 0$ the equalities below are valid

$$H_t = \sum_{n,k=1}^N \mathbf{I}_{\{0\}} (G_k(t) - G_n(t)) X_t^k e_n = K(t) X_t = K(t) X_{t-} \ \ \mathcal{P} - \text{a.s.}, \tag{12}$$

where $K(t)$ is the $N \times N$-dimensional matrix with the components (4).

The function $K(t)$ has the following properties.

1. $K(t) \equiv K^\top(t)$ for any $t \geqslant 0$.
2. The number of $K(\cdot)$ jumps occurred in any finite time interval is finite due to condition 4.
3. $K(t)$ is not a *cádlág*-function [25].
4. $\mathcal{P}\{\|\Delta K(t)\|\|\Delta X_t\| > 0\} = 0$ for any $t \geqslant 0$.
5. For any $t \geqslant 0$ there exists a transformation $T(t)$ such that the matrix $T(t)K(t)$ is trapezoid with orthogonal strings and 0 and 1 as the components.
6. $\mathcal{P}\{T(t)H_t \in \mathbb{S}^N\} = 1$ for any $t \geqslant 0$.

Let us define a $\mathcal{Y}_{t+}$-adapted process $V_t = \operatorname{col}(V_t^1, \ldots, V_t^N)$ with the *cádlág*-trajectories:

$$V_t \triangleq T(t+) H_{t+}. \tag{13}$$

From (12) and (13) it follows that $V_t = J(t) X_t$ $\mathcal{P}$-a.s., where $J(t) \triangleq T(t+)K(t+)$.

We denote the set of the process $V$ discontinuity by $\mathcal{V}$, $\mathcal{X}$ stands for the set of $X$ discontinuity and $\mathcal{J}$ for the analogous set of the process $J$. The sets $\mathcal{V}$ and $\mathcal{X}$ are random, in contrast $\mathcal{J}$ is nonrandom. The process $V_t$ is purely discontinuous, and due to property 4 it can be rewritten in the form

$$V_t = J(0)X_0 + \sum_{\kappa \in \mathcal{V}:\, \kappa \leqslant t} \Delta V_\kappa = J(0)X_0 + \sum_{\kappa \in \mathcal{J}:\, \kappa \leqslant t} \Delta J(\kappa) X_\kappa + \sum_{\kappa \in \mathcal{V} \backslash \mathcal{J}:\, \kappa \leqslant t} J(\kappa) \Delta X_\kappa =$$

$$= J(0)X_0 + \underbrace{\sum_{\kappa \in \mathcal{J}:\, \kappa \leqslant t} \Delta J(\kappa) X_\kappa}_{} + \sum_{\kappa \in \mathcal{X}:\, \kappa \leqslant t} J(\kappa) \Delta X_\kappa = J(0)X_0 + \underbrace{\sum_{\kappa \in \mathcal{J}:\, \kappa \leqslant t} \Delta J(\kappa) X_\kappa}_{\triangleq D_t} + \underbrace{\int_0^t J(s) dX_s}_{\triangleq R_t}. \tag{14}$$

Due to the definition $V_t \in \mathbb{S}^N$ for $\forall\, t \geqslant 0$. The process $D_t$ characterizes the observable jumps at the nonrandom moments caused by $J(t)$ changes, and $R_t$ is an observable part of the state $X_t$ jumps, occurred, at some random instants.

As a second part of the transformed observations, we choose the $N$-dimensional random process $C_t \triangleq \mathrm{col}(C_t^1, \dots, C_t^N)$: the components $C_t^n$ count the jumps of the process $V_t$ into the state $e_n$, occurred *at the random instants* over the interval $[0, t]$:

$$C_t^n = \int_0^t (1 - e_n^\top V_{s-}) e_n^\top dR_s. \tag{15}$$

The third part of the transformed observations is the $N$-dimensional process $D_t$ with the jumps at the nonrandom moments.

**Lemma 1.** *If $\overline{\mathcal{Y}}_t \triangleq \sigma\{(U_s, C_s, D_s), \ s \in [0, t]\}$, then the coincidence $\overline{\mathcal{Y}}_t = \mathcal{Y}_{t+}$ is true for any $t \geqslant 0$.*

Correctness of the Lemma assertion follows immediately from the fact the composite process $(U_t, C_t, D_t)$ is constructed to be $\mathcal{Y}_{t+}$-adapted, and one-to-one correspondence of the $(U, C, D)$ and $Y$ paths:

$$\begin{cases} U_t = \int_{0_t}^t \left( \frac{d\langle Y, Y \rangle_u}{du} \Big|_{u=s+} \right)^{-1/2} dY_s, \\ C_t = \int_0^t (I - \mathrm{diag}\, V_{s-}) dV_s - \sum_{\kappa \in \mathcal{J}: \kappa \leqslant t} (I - \mathrm{diag}\, V_{\kappa-}) \Delta V_\kappa, \\ D_t = \sum_{\kappa \in \mathcal{J}: \kappa \leqslant t} (I - \mathrm{diag}\, V_{\kappa-}) \Delta V_\kappa, \\ V_t = T(t+) H_{t+}, \\ H_t \triangleq \sum_{n=1}^N \mathbf{I}_{\{0\}} \left( \frac{d\langle Y, Y \rangle_s}{ds} \Big|_{s=t-} - G_n(t) \right) e_n, \end{cases} \tag{16}$$

$$\begin{cases} V_t = D_t + \int_0^t \sum_{(i,j):\, i \neq j}^N V_{s-}^i (e_j - e_i) dC_s^j. \\ Y_t = \int_0^t \sum_{n=1}^N V_s^n G_n^{1/2}(s) dU_s, \end{cases} \tag{17}$$

Below we use the following notations: $\mathbf{1}$ is a row vector of the appropriate dimensionality formed by units, $J_n(s) \triangleq e_n^\top J(s)$ is the $n$-th row of the matrix $J(s)$,

$$\Gamma_n(s) \triangleq \mathrm{diag}(J_n(s)) \Lambda^\top(s)(I - \mathrm{diag}\, J_n(s)). \tag{18}$$

**Lemma 2.** *The process $C_t = \mathrm{col}(C_t^1, \dots, C_t^N)$ has the following properties.*

1. *$n$-th component $C_t^n$ allows the martingale representation*

$$C_t^n = \int_0^t \mathbf{1}\Gamma_n(s) X_s ds + \int_0^t (1 - J_n(s) X_{s-}) J_n(s) dM_s^X.$$

2. *$[C^n, C^m]_t \equiv 0$ for any $n \neq m$;*

$$\langle C^n, C^n \rangle_t = \int_0^t \mathbf{1}\Gamma_n(s) X_s ds. \tag{19}$$

3. *The innovation processes*

$$v_t^n \triangleq \int_0^t \left( dC_s^n - \mathbf{1}\Gamma_n(s) \widehat{X}_s ds \right), \quad n = \overline{1, N} \tag{20}$$

*are $\overline{\mathcal{Y}}_t$-adapted martingales with the quadratic characteristics*

$$\langle v^n, v^n \rangle_t = \int_0^t \mathbf{1}\Gamma_n(s) \widehat{X}_s ds. \tag{21}$$

Proof of Lemma 2 is given in Appendix A.

Finally, the transformed observations $(U, C, D)$ take the form

$$
\begin{cases}
U_t = \int_0^t \overline{f}(s) X_s ds + \overline{W}_t, \\
C_t^n = \int_0^t \mathbf{1}\Gamma_n(s) X_s ds + \int_0^t (1 - J_n(s) X_{s-}) J_n(s) dM_s^X, \ n = \overline{1, N}, \\
D_t = J(0) X_0 + \sum_{\kappa \in \mathcal{J}:\, \kappa \leqslant t} \Delta J(\kappa) X_\kappa.
\end{cases} \tag{22}
$$

**Theorem 1.** *The optimal filtering estimate $\widehat{X}_t$ is a strong solution to the SDS*

$$
\widehat{X}_t = \left( (D_0)^\top J(0) \pi_0 \right)^+ \operatorname{diag}(D_0) J(0) \pi_0 + \int_0^t \Lambda^\top(s) \widehat{X}_s ds + \int_0^t \left( \operatorname{diag} \widehat{X}_s - \widehat{X}_s \widehat{X}_s^\top \right) \overline{f}^\top(s) d\omega_s +
$$

$$
+ \sum_{n=1}^N \int_0^t \left( \Gamma_n(s) - \mathbf{1}\Gamma_n(s) \widehat{X}_{s-} I \right) \widehat{X}_{s-} \left( \mathbf{1}\Gamma_n(s) \widehat{X}_{s-} \right)^+ d\nu_s^n +
$$

$$
+ \sum_{\kappa \in \mathcal{J}:\, \kappa \leqslant t} \left( \left( \Delta D_\kappa^\top \Delta J(\kappa) \widehat{X}_{\kappa-} \right)^+ \operatorname{diag}(\Delta D_\kappa) \Delta J(\kappa) - I \right) \widehat{X}_{\kappa-}, \tag{23}
$$

*where*

$$
\omega_t \triangleq U_t - \int_0^t \overline{f}(s) \widehat{X}_s ds \tag{24}
$$

*and $A^+$ is a Moore–Penrose pseudoinverse. The solution is unique within the class of nonnegative piecewise-continuous $\mathcal{Y}_{t+}$-adapted processes with discontinuity set lying in $\mathcal{V}$.*

Proof of Theorem 1 is given in Appendix B.

The transformed observations (22) along with Theorem 1 prompt a condition of the *exact* identifiability of the state $X_t$ given indirect noisy observations $Y_t$ (2).

**Corollary 1.** *If for any $n \neq m$ ($n, m = \overline{1, N}$) the inequalities $G_n(s) \neq G_m(s)$ are true almost everywhere on $[0, t]$, then $\widehat{X}_t = X_t$ $\mathcal{P}$-a.s., and $X_t$ is the solution to SDS (23).*

The proof of Corollary 1 is given in Appendix C.

## 4. Numerical Algorithms of Optimal Filtering

### 4.1. Optimal Filtering Given Discretized Observations

The latter section contains the stochastic system (23) defining the optimal filtering estimate $\widehat{X}_t$. The problem of its numerical realization seems routine: we should apply the corresponding methods of numerical integration of SDS with jumps on the RHS [26]. However, this simplicity is illusory. The problem is that the "new" countable observation $C_t$ and discrete-time one $D_t$ are results of certain transform of the available observation $Y$, and this transform includes a limit passage operation. In fact, to obtain $C_t$ we have to estimate/restore the current value of the derivative $\frac{d\langle Y, Y \rangle_{t+}}{dt}$. First, this leads to some time delay to accumulate observations $Y_t$. Second, any pre-limit variant of $C_t$ either has a.s. continuous trajectories or represents their sampling, which demonstrates oscillating nature. Third, the considered filtering estimate is the CME of the state $X_t$ given the observations $Y$ up to the moment $t$. The CME has natural properties: its components are a.s. non-negative and satisfy the normalization condition. The estimates and approximations having these properties are referred in the paper as the stable ones. Mostly, the conventional numerical algorithms do not provide these properties for the calculated approximations. They can preserve the normalization condition only, but the components can have the arbitrary signs and absolute values.

In the paper we present another approach to the numerical realization of the filtering algorithm above. We discretize the available observations $Y$ by time with the increment $h$ and then solve

the optimal state filtering problem given discretized observations. The estimate can be considered as approximation of the one given the initial continuous-time observations. Properties of the CME guarantee the stability of the proposed approximation.

To simplify derivation of the numerical algorithm and its accuracy analysis we investigate the time-invariant subset of the observation system (1), (2), i.e., $\Lambda(t) \equiv \Lambda$, $A(t) \equiv A$, $G_n(t) \equiv G_n$, $n = \overline{1, N}$. The observations are discretized with the time increment $h$:

$$Y_r \triangleq \int_{t_{r-1}}^{t_r} f X_s ds + \int_{t_{r-1}}^{t_r} \sum_{n=1}^{N} X_s^n G_n^{1/2} dW_s, \quad r \in \mathbb{N}, \tag{25}$$

where $t_r \triangleq rh$ are equidistant time instants. We denote $\mathfrak{Y}_r \triangleq \sigma\{Y_s : 1 \leqslant s \leqslant r\}$ non-decreasing collection of $\sigma$-algebras generated by the time-discretized observations; $\mathfrak{Y}_0 \triangleq \{\varnothing, \Omega\}$.

*The optimal state filtering problem given discretized observations* is to find $\widehat{X}_r \triangleq \mathbf{E}\{X_{t_r} | \mathfrak{Y}_r\}$.

Let us consider asymptotics of $\widehat{X}$. We fix some $T > 0$ and consider a condensed sequence of binary meshes $\{\frac{rT}{2^n}\}_{r=\overline{1,2^n}}$ with time increments $h_n \triangleq \frac{T}{2^n}$ and corresponding increasing sequence of $\sigma$-subalgebras $\{\mathfrak{Y}_{2^n}^n\}$: $\mathfrak{Y}_{2^n}^n \triangleq \sigma\{Y_r, 1 \leqslant r \leqslant 2^n\}$. The observation process $\{Y_t\}$ is separable, hence $\sigma\{\bigcup_{n=1}^{\infty} \mathfrak{Y}_n\} = \mathcal{Y}_T$. Then, by Levy theorem $\widehat{X}_{2^n} \triangleq \mathbf{E}\{X_T | \mathfrak{Y}_n\} \xrightarrow{n \to \infty} \mathbf{E}\{X_T | \mathcal{Y}_T\} = \mathbf{E}\{X_T | \mathcal{Y}_{T+}\} \triangleq \widehat{X}_T$ $\mathcal{P}$-a.s. Moreover, since $\mathbf{E}\{\widehat{X}_T\} \equiv \mathbf{E}\{\widehat{X}_{2^n}\} = \pi(T)$, the $\mathcal{L}_1$-convergence is also true: $\lim_{n \to \infty} \mathbf{E}\{|\widehat{X}_T - \widehat{X}_{2^n}|\} = 0$. The convergence also holds, if we replace the sequence of the binary meshes by any condensed sequence with vanishing step. So, we can conclude that the optimal filtering given the discretized observation is a way to design the stable convergent approximations without observation transform $Y \to (U, C, D)$ introduced in the previous section.

To derive the filtering formula we use the approach of [27] and the mathematical induction.

In the case $r = 0$ we have

$$\widehat{X}_0 = \mathbf{E}\{X_0 | \mathfrak{Y}_0\} = \mathbf{E}\{X_0\} = \pi. \tag{26}$$

Let for some $r \in \mathbb{N}$ the estimate $\widehat{X}_{r-1} = \mathbf{E}\{X_{t_{r-1}} | \mathfrak{Y}_{r-1}\}$ be known. Now we calculate $\widehat{X}_r$ at the next time instant. To do this we have to specify the mutual conditional distribution $(X_{t_r}, Y_r)$ with respect to $\mathfrak{Y}_{r-1}$. From the observation model and ([10] Lemma 7.5) it follows that the conditional distribution of $Y_r$ given $\sigma$-algebra $\mathcal{F}_{t_r}^X \vee \mathfrak{Y}_{r-1}$ is Gaussian with the parameters

$$\mathbf{E}\{Y_r | \mathcal{F}_{t_r}^X\} = fv_r, \qquad \mathrm{cov}(Y_r, Y_r | \mathcal{F}_{t_r}^X) = \sum_{n=1}^{N} v_r^n G_n. \tag{27}$$

Here, $v_r = \mathrm{col}(v_r^1, \ldots, v_r^N) \triangleq \int_{t_{r-1}}^{t_r} X_s ds$ is a random vector composed of the occupation times of the process $X$ in each state $e_n$ during the interval $[t_{r-1}, t_r]$.

Below in the presentation we use the following notations:

- $\mathcal{D} \triangleq \{u = \mathrm{col}(u^1, \ldots, u^N) : u^n \geqslant 0, \sum_{n=1}^{N} u^n = h\}$ is an $(N-1)$-dimensional simplex in the space $\mathbb{R}^M$; $\mathcal{D}$ is a distribution support of the vector $v_r$;
- $\Pi \triangleq \{\pi = \mathrm{col}(\pi^1, \ldots, \pi^N) : \pi^n \geqslant 0, \sum_{n=1}^{N} \pi^n = 1\}$ is a "probabilistic simplex" formed by the possible values of $\pi$;
- $N_r^X$ is a random number of the state $X_t$ transitions, occurred on the interval $[t_{r-1}, t_r]$,
- $a_r^s \triangleq \{\omega \in \Omega : N_r^X(\omega) \leqslant s\}$, $A_r^s \triangleq \prod_{q=1}^{r} a_q^s$;
- $\rho^{k,\ell,q}(du)$ is a conditional distribution of the vector $X_{t_r}^\ell \mathbf{I}_{\{q\}}(N_r^X) v_r$ given $X_{t_{r-1}} = e_k$, i.e., for any $\mathcal{G} \in \mathcal{B}(\mathbb{R}^M)$ the following equality is true:

$$\mathbf{E}\left\{\mathbf{I}_{\mathcal{G}}(v_r) \mathbf{I}_{\{q\}}(N_r^X) X_{t_r}^\ell | X_{t_{r-1}} = e_k\right\} = \int_{\mathcal{G}} \rho^{k,\ell,q}(du);$$

- $\mathcal{N}(y, m, K) \triangleq (2\pi)^{-M/2}\det^{-1/2}K\exp\left\{-\frac{1}{2}\|y-m)\|^2_{K^{-1}}\right\}$ is an $M$-dimensional Gaussian *probability density function* (pdf) with the expectation $m$ and nondegenerate covariance matrix $K$;
- $\|\alpha\|^2_K \triangleq \alpha^\top K\alpha, \quad \langle\alpha, \beta\rangle_K \triangleq \alpha^\top K\beta.$

Markovianity of $\{(X_{t_r}, Y_r)\}_{r\geqslant 0}$, formula of the total probability and Fubini theorem provide the equalities below for any set $\mathcal{A} \in \mathcal{B}(\mathbb{R}^M)$

$$\mathbf{E}\left\{X_{t_r}\mathbf{I}_\mathcal{A}(Y_r)\big|\mathfrak{Y}_{r-1}\right\} = \mathbf{E}\left\{\mathbf{E}\left\{X_{t_r}\mathbf{I}_\mathcal{A}(Y_r)\big|\mathcal{F}^X_{t_r}\vee\mathfrak{Y}_{r-1}\right\}\big|\mathfrak{Y}_{r-1}\right\} =$$

$$= \mathbf{E}\left\{X_{t_r}\int_\mathcal{A}\mathcal{N}\left(y, fv_r, \sum_{p=1}^N v^p_r G_p\right)dy\Big|\mathfrak{Y}_{r-1}\right\} =$$

$$= \mathbf{E}\left\{\mathbf{E}\left\{X_{t_r}\int_\mathcal{A}\mathcal{N}\left(y, fv_r, \sum_{p=1}^N v^p_r G_p\right)dy\Big|X_{t_{r-1}}\vee\mathfrak{Y}_{r-1}\right\}\Big|\mathfrak{Y}_{r-1}\right\} =$$

$$= \mathbf{E}\left\{\sum_{\ell=1}^N e_\ell\sum_{q=0}^\infty\sum_{k=1}^N e_k^\top X_{t_{r-1}}\int_\mathcal{D}\int_\mathcal{A}\mathcal{N}\left(y, fu, \sum_{p=1}^N u^p G_p\right)dy\rho^{k,\ell,q}(du)\Big|\mathfrak{Y}_{r-1}\right\} =$$

$$= \sum_{\ell=1}^N e_\ell\int_\mathcal{A}\left[\sum_{k=1}^N \widehat{X}^k_{r-1}\sum_{q=0}^\infty\int_\mathcal{D}\mathcal{N}\left(y, fu, \sum_{p=1}^N u^p G_p\right)\rho^{k,\ell,q}(du)\right]dy.$$

This means that the integrand in the square brackets defines the conditional distribution $(X_{t_r}, Y_r)$ given $\mathfrak{Y}_{r-1}$. Further, the conditional distribution $\widehat{X}_r$ is defined component-wisely by the generalized Bayes rule [10]

$$\widehat{X}^j_r = \frac{\sum_{k=1}^N \widehat{X}^k_{r-1}\sum_{q=0}^\infty\int_\mathcal{D}\mathcal{N}\left(Y_r, fu, \sum_{p=1}^N u^p G_p\right)\rho^{k,j,q}(du)}{\sum_{i,\ell=1}^N \widehat{X}^i_{r-1}\sum_{c=0}^\infty\int_\mathcal{D}\mathcal{N}\left(Y_r, fv, \sum_{n=1}^N v^n G_n\right)\rho^{i,\ell,c}(dv)}, \quad j = \overline{1, N}. \tag{28}$$

So, we have proved the following

**Lemma 3.** *If for the observation system* (1), (2) *conditions 1–3 are valid, then the filtering estimate* $\widehat{X}_r$ *given the discretized observations is defined by* (26) *at* $r = 0$, *and by recursion* (28) *at the instant* $t_r$ *of the discretized observation* $Y_r$ *reception.*

*4.2. Stable Analytic Approximations*

Recursion (23) cannot be realized directly because of infinite summation both in the numerator and denominator. We replace them by the finite sums, and the corresponding vector sequence $\overline{X}_r(s)$, calculated by the formula

$$\overline{X}^j_r(s) = \frac{\sum_{k=1}^N \overline{X}^k_{r-1}(s)\sum_{q=0}^s\int_\mathcal{D}\mathcal{N}\left(Y_r, fu, \sum_{p=1}^N u^p G_p\right)\rho^{k,j,q}(du)}{\sum_{i,\ell=1}^N \overline{X}^i_{r-1}(s)\sum_{c=0}^s\int_\mathcal{D}\mathcal{N}\left(Y_r, fv, \sum_{n=1}^N v^n G_n\right)\rho^{i,\ell,c}(dv)}, \quad j = \overline{1, N} \tag{29}$$

is called *the analytic approximation of the s-th order* of $\widehat{X}_r$. Obviously, that $\overline{X}_r(s)$ is stable.

Let us introduce the following positive random numbers and matrices:

$$\xi^{kj}_q \triangleq \sum_{m=0}^s\int_\mathcal{D}\mathcal{N}\left(Y_q, fu, \sum_{p=1}^N u^p G_p\right)\rho^{k,j,m}(du),$$

$$\theta^{kj}_q \triangleq \sum_{m=s+1}^\infty\int_\mathcal{D}\mathcal{N}\left(Y_q, fu, \sum_{p=1}^N u^p G_p\right)\rho^{k,j,m}(du), \tag{30}$$

$$\xi_q \triangleq \|\xi^{kj}_q\|_{k,j=\overline{1,N}}, \qquad \theta_q \triangleq \|\theta^{kj}_q\|_{k,j=\overline{1,N}}.$$

The estimates $\widehat{\mathsf{X}}_r$ (28) and $\overline{\mathsf{X}}_r(s)$ (29) can be rewritten in the recurrent form:

$$\widehat{\mathsf{X}}_r = (\mathbf{1}(\xi_r + \theta_r)^\top \widehat{\mathsf{X}}_{r-1})^{-1}(\xi_r + \theta_r)^\top \widehat{\mathsf{X}}_{r-1}, \tag{31}$$

$$\overline{\mathsf{X}}_r(s) = (\mathbf{1}\zeta_r^\top \overline{\mathsf{X}}_{r-1}(s))^{-1}\zeta_r^\top \overline{\mathsf{X}}_{r-1}(s). \tag{32}$$

Let us define the global distance [28] between the estimates $\{\overline{\mathsf{X}}_r(s)\}$ and $\{\widehat{\mathsf{X}}_r\}$ as

$$\Sigma_r(s) \triangleq \sup_{\pi \in \Pi} \mathbf{E}\left\{\|\widehat{\mathsf{X}}_r - \overline{\mathsf{X}}_r(s)\|_1\right\} = \sup_{\pi \in \Pi}\sum_{j=1}^{N} \mathbf{E}\left\{|\widehat{\mathsf{X}}_r^j - \overline{\mathsf{X}}_r^j(s)|\right\}. \tag{33}$$

The pretty natural characteristic shows the maximal expected divergence of the recursions (28) and (29) at the $r$-th step.

The assertion below defines an upper bound of the characteristic $\Sigma_r(s)$.

**Lemma 4.** *If the conditions of Lemma 3 are valid, then*

$$\Sigma_r(s) \leqslant 2 - 2\left(1 - C_1 \frac{(\overline{\lambda}h)^{s+1}}{(s+1)!}\right)^r, \tag{34}$$

*where $\overline{\lambda} \triangleq \max_{1 \leqslant n \leqslant N} |\lambda_{nn}|$, and $C_1 = C_1(h, \overline{\lambda}) \in (0,1)$ is the following parameter:*

$$C_1 \triangleq e^{-\overline{\lambda}h} \frac{(s+1)!}{(\overline{\lambda}h)^{s+1}} \sum_{k=s+1}^{\infty} \frac{(\overline{\lambda}h)^k}{k!}, \tag{35}$$

*which is bounded from above: $C_1 \frac{(\overline{\lambda}h)^{s+1}}{(s+1)!} < 1$.*

The proof of Lemma 4 is given in Appendix D.

Assertion of Lemma brings the practical benefit. The Lemma does not contain any asymptotic requirements neither to the approximation order $s$ nor to the discretization step $h$: inequality (34) is universal. Mostly, in the digital control systems the data acquisition rate is fixed or bounded from above. There are some extra algorithmic limitations of the rate: the "raw" data should be preprocessed, smoothed, averaged, refined from outliers, etc. For example, utilization of the central limit theorem [29] and diffusion approximation framework [30] for the the renewal processes is legitimate with significant averaging intervals, and their length depends on the process moments.

Now we fix the time instant $T$ and consider an asymptotic $h \to 0$. In this case $r = \frac{T}{h} \to \infty$ and

$$\Sigma_{\frac{T}{h}}(s) \leqslant 2 - 2\left(1 - C_1 \frac{(\overline{\lambda}h)^{s+1}}{(s+1)!}\right)^{\frac{T}{h}} \sim 2\overline{\lambda}T\frac{(\overline{\lambda}h)^s}{(s+1)!}.$$

### 4.3. Stable Numerical Approximations

In the recursion (32) we use the integrals $\zeta_r^{ij}$, which cannot be calculated analytically. The numerical integration brings some extra approximation error. Let us investigate its affect to the total accuracy of the filter numerical realization.

The integrals $\xi^{ij}(y)$ are usually approximated by the sums

$$\xi^{ij}(y) \approx \psi^{ij}(y) \triangleq \sum_{\ell=1}^{L} \mathcal{N}\left(y, fw_\ell, \sum_{p=1}^{N} w_\ell^p g_p\right)\varrho_\ell^{ij}, \qquad \psi(y) \triangleq \|\psi^{ij}(y)\|_{i,j=\overline{1,N}}, \tag{36}$$

which are defined by the collection of the pairs $\{(w_\ell, \varrho_\ell^{ij})\}_{\ell=\overline{1,L}}$. Here, $w_\ell \triangleq \mathrm{col}(w_\ell^1, \ldots, w_\ell^N) \in \mathcal{D}$ are the points, and $\varrho_\ell^{ij} \geqslant 0$ ($\ell = \overline{1,L}$) are the weights: $\sum_{j=1}^{N}\sum_{\ell=1}^{L} \varrho_\ell^{ij} \leqslant Q \leqslant 1$.

In complete analogy with $\xi_q$ we define the approximations $\psi_q \triangleq \|\psi^{ij}(Y_q)\|_{i,j=\overline{1,N}}$. By construction, the elements of $\psi_q$ are positive random values, hence the approximation $\widetilde{X}_r$

$$\widetilde{X}_r \triangleq (\mathbf{1}\psi_r^\top \widetilde{X}_{r-1})^{-1}\psi_r^\top \widetilde{X}_{r-1}, \quad \widetilde{X}_0 = \pi \tag{37}$$

is stable. Below we denote the numerical integration errors and their absolute values as follows

$$\gamma^{kj} \triangleq \psi^{kj} - \xi^{kj}, \qquad \gamma_r \triangleq \|\gamma^{kj}(Y_r)\|_{k,j=\overline{1,N}} \tag{38}$$

$$\overline{\gamma}^{kj} \triangleq |\gamma^{kj}|, \qquad \overline{\gamma}_r \triangleq \left\||\gamma^{kj}(Y_r)|\right\|_{k,j=\overline{1,N}}. \tag{39}$$

So, the recursion (32) is replaced by the scheme (37), holding the common initial condition $\pi$.

Both (32) and (37) are constructed in light of the event $A_r^s$: the state transition numbers do not exceed the threshold $s$ over any subintervals $[t_{q-1}, t_q]$ belonging to $[0, t_r]$. So, the distance between $\widetilde{X}_r$ and $\overline{X}_r(s)$ should be determined taking into account $A_r^s$. In view of this fact, we propose the pseudo-metrics

$$\mathcal{E}_r(s) \triangleq \sup_{\pi \in \Pi} \mathbf{E}\left\{\mathbf{I}_{A_r^s}(\omega)\|\widetilde{X}_r - \overline{X}_r(s)\|_1\right\} = \sup_{\pi \in \Pi} \sum_{n=1}^N \mathbf{E}\left\{\mathbf{I}_{A_r^s}(\omega)|\widetilde{X}_r^n - \overline{X}_r^n(s)|\right\}. \tag{40}$$

This index reflects maximal divergence of the algorithms (32) and (37) after $r$ steps, being started from the arbitrary but common initial condition.

**Theorem 2.** *If the inequality*

$$\max_{i=\overline{1,N}} \sum_{j=1}^N \int_{\mathbb{R}^M} |\psi^{ij}(y) - \xi^{ij}(y)|dy < \delta \tag{41}$$

*is true for the numerical integration scheme (36), then the distance $\mathcal{E}_r(s)$ is bounded from above:*

$$\mathcal{E}_r(s) \leqslant 2rQ^{r-1}\delta. \tag{42}$$

The proof of Theorem 2 is given in Appendix E.

The chance to describe the accuracy of the numerical algorithm for the stochastic filtering using only the condition (41), related to the calculus, looks remarkable. Furthermore, if the total weight $Q = \sum_{\ell,j} \varrho_\ell^{ij}$ separates from the unity, i.e., $Q < 1$, then the index $\mathcal{E}_r(s)$ is a *sublinear* function of $r$, so as the index $\Sigma_r(s)$ of the analytic accuracy is. Notably, that in the classic numerical algorithms of the SDS solution the global error grows *linearly* with respect to the number of steps $r$ [26].

The precision characteristics of both the analytical approximation and its numerical realization should be aggregated into the one. If the conditions of Lemma 4 and Theorem 2 are valid, then the local distance (i.e., the distance after one iteration) between the optimal filtering estimate and its numerical approximation can be bounded from above:

$$\tau(s) \triangleq \sup_{\pi \in \Pi} \mathbf{E}\left\{\|\widehat{X}_1 - \widetilde{X}_1\|_1\right\} \leqslant \sup_{\pi \in \Pi} \mathbf{E}\left\{\mathbf{I}_{a_1^s}(\omega)\|\widetilde{X}_1 - \overline{X}_1(s) + \overline{X}_1(s) - \widehat{X}_1\|_1 + \mathbf{I}_{\overline{a}_1^s}(\omega)\|\widetilde{X}_1 - \overline{X}_1(s)\|_1\right\} \leqslant$$

$$\leqslant 2\mathcal{P}\left\{\overline{a}_1^s\right\} + \sup_{\pi \in \Pi} \mathbf{E}\left\{\|\overline{X}_1(s) - \widehat{X}_1\|_1\right\} + \sup_{\pi \in \Pi} \mathbf{E}\left\{\mathbf{I}_{a_1^s}(\omega)\|\widetilde{X}_1 - \overline{X}_1(s)\|_1\right\} =$$

$$= 2\mathcal{P}\left\{\overline{a}_1^s\right\} + \sigma(s) + \mathcal{E}_1(s) \leqslant 4\frac{(\overline{\lambda}h)^{s+1}}{(s+1)!} + 2\delta. \tag{43}$$

The global distance between $\widehat{X}_r \triangleq \mathbf{E}\{X_r|\mathfrak{Y}_r\}$ and $\widetilde{X}_r$ can be bounded in the similar way:

$$\mathcal{T}(s) \triangleq \sup_{\pi \in \Pi} \mathbf{E}\left\{\|\widehat{X}_r - \widetilde{X}_r\|_1\right\} \leqslant 4\left[1 - \left(1 - \frac{(\overline{\lambda}h)^{s+1}}{(s+1)!}\right)^r\right] + 2rQ^{r-1}\delta. \tag{44}$$

We could choose the parameters $(h, s)$ of the analytical approximation and $\delta$ of the numerical integration independently each other. However, both the limitation of the computational resources and the accuracy requirements lead to the necessity of the mutual optimization of $(h, s, \delta)$.

Let us fix some time horizon $T$ along with the order $s$ of analytical approximation, and consider the asymptotic $r \to \infty$, or, equivalently, $h = \frac{T}{r} \to 0$. Due to the Bernoulli inequality, and condition $0 < Q \leqslant 1$ we have that

$$
\sup_{\pi \in \Pi} \mathbf{E} \left\{ \| \widetilde{X}_{T/h} - \widehat{X}_{T/h} \|_1 \right\} \leqslant 4 \left[ 1 - \left( 1 - \frac{(\overline{\lambda}h)^{s+1}}{(s+1)!} \right)^r \right] + 2rQ^{r-1}\delta \leqslant 4r \frac{(\overline{\lambda}h)^{s+1}}{(s+1)!} + 2rQ^{r-1}\delta =
$$
$$
= 4\overline{\lambda}T \frac{(\overline{\lambda}h)^s}{(s+1)!} + 2rQ^{r-1}\delta \leqslant 2T \left( 2\overline{\lambda} \frac{(\overline{\lambda}h)^s}{(s+1)!} + \frac{\delta}{h} \right). \quad (45)
$$

The first summand in the brackets represents the contribution of the analytical approximation error, the second one reflects the error of the specified numerical integration scheme. Obviously, the optimal choice of the parameters provides an equal infinitesimal order for both the summands, and it is possible when $\delta \sim \frac{(\overline{\lambda}h)^{s+1}}{\overline{\lambda}}$.

### 4.4. Numerical Example

To illustrate the correspondence between the theoretical estimate and its realization along with the performance of the numerical algorithm, we consider the filtering problem for the observation system (1) and (2) with the following parameters: $t \in [0, 1]$, $N = 3$,

$$
\Lambda = \begin{bmatrix} -1.0 & 0.2 & 0.8 \\ 0.8 & -1.0 & 0.2 \\ 0.2 & 0.8 & -1.0 \end{bmatrix}, \quad \pi = \begin{bmatrix} 0.333 \\ 0.333 \\ 0.334 \end{bmatrix}, \quad f = \begin{bmatrix} 0.0 \\ 0.0 \\ 0.0 \end{bmatrix}, \quad \begin{array}{l} G_1 = 1.0, \\ G_2 = 4.0, \\ G_3 = 9.0. \end{array}
$$

The specified observation system is the one with state-dependent noise, and the conditions of Corollary 1 hold, so the optimal filter (23) restores the MJP state precisely under available noisy observations. Let us verify this theoretical fact, using the recursive algorithm (37). We choose the analytical approximation of the order $s = 1$ with numerical integration by the simple midpoint rectangle scheme and calculate estimate approximations with decreasing time-discretization step: $h = 0.01; 0.001; 0.0001; 0.00001$. We expect the descent of the estimation error characterized by the MS-criterion $\mathcal{S}_t(h) = \sqrt{\mathbf{E} \left\{ \| X_t - \widetilde{X}_{\frac{t}{h}} \|_2^2 \right\}}$. To calculate the criterion, we use the Monte–Carlo method over the test sample of the size 1000. Figure 1 presents the corresponding plots of the quality index $\mathcal{S}_t(h)$ for various values of $h$.

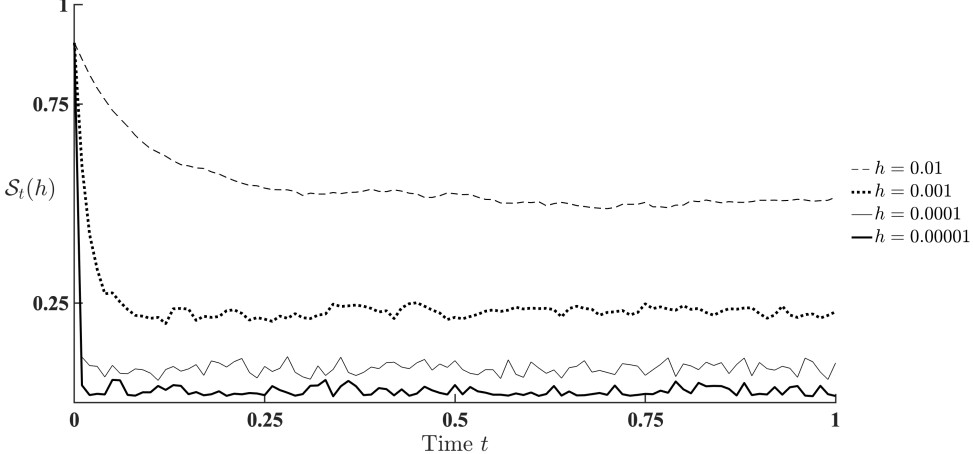

**Figure 1.** Estimation quality index $\mathcal{S}_t(h)$ depending on the time-discretization step $h$.

The determination of the precision order provided by the chosen numerical integration method is out of the scope of this investigation. Nevertheless, one can see the expected decrease of the estimation error when the time-discretization step descends. We appraise this result as a practical confirmation of both the theoretical assertions and numerical algorithm.

## 5. Conclusions

In this paper, we investigated the optimal filtering problem of the MJP states, given the indirect noisy continuous-time observations. The observation noise intensity was a function of the estimated state, so it was impossible to apply the classic Wonham filter to this observation system. To overcome this obstacle, we suggested an observation transform. On the one hand, the transformed observations remained to be equivalent to the original one from the informational point of view. On the other hand, the "new" observations allowed to apply the effective stochastic analysis framework to process them. We derived the optimal filtering estimate theoretically as a unique strong solution to some discrete–continuous stochastic differential system. The transformed observations included derivative of the quadratic characteristics, i.e., the result of some limit passage in the stochastic settings. Hence, the subsequent numerical realization of the filtering became challenging. We proposed to approximate the initial continuous-time filtering problem by a sequence of the optimal ones given the time-discretized observations. We also involved numerical integration schemes to calculate the integrals included in the estimation formula. We prove assertions, characterizing the accuracy of the numerical approximation of the filtering estimate, i.e., the distance between the calculated approximation and optimal discrete-time filtering estimate. The accuracy depended on the observation system parameters, time discretization step, a threshold of state transition number during the time step, and the chosen scheme of the numerical integration. We suggested the whole class of numerical filtering algorithms. In each case, one could choose any specific algorithm individually, taking into account characteristics of the concrete observation system, accuracy requirements, and available computing resources.

We do not consider the presented investigations as completed. First, the characterization of the distance between the initial optimal continuous-time filtering estimate and its proposed approximation is still an open problem. Second, we can use the theoretical solution to the MJP filtering problem as a base of numerical schemes for the diffusion process filtering, given the observations with state-dependent noise. Third, the obtained optimal filtering estimate looks a springboard for a solution to the optimal stochastic control of the Markov jump processes, given both the counting and diffusion observations with state-dependent noise. All of this research is in progress.

**Author Contributions:** Conceptualization, A.B., I.S.; methodology, A.B.; formal analysis and investigation, A.B., I.S.; writing—original draft preparation, A.B.; writing—review and editing, I.S.; supervision, I.S. All authors have read and agreed to the published version of the manuscript.

**Funding:** This research received no external funding.

**Conflicts of Interest:** The authors declare no conflict of interest.

## Abbreviations

The following abbreviations are used in this manuscript:

| | |
|---|---|
| CME | Conditional mathematical expectation |
| MJP | Markov jump process |
| pdf | Probability density function |
| RHS | Right-hand side |
| SDS | Stochastic differential system |

## Appendix A. Proof of Lemma 2

From (14), (15), the identity $\operatorname{diag}(a)b \equiv \operatorname{diag}(b)a$, the fact that $J_n(t) \neq J_n(t-)$ at most at finite points of any finite interval and property 4 of the function $K(t)$, the following equalities are true

$$
\begin{aligned}
C_t^n &= \int_0^t (1 - e_n^\top V_{s-}) e_n^\top dR_s = \int_0^t (1 - e_n^\top V_{s-}) e_n^\top J(s)(\Lambda^\top(s) X_{s-} ds + dM_s^X) = \\
&= \int_0^t (1 - J_n(s-) X_{s-}) J_n(s-) \Lambda^\top(s) X_{s-} ds + \int_0^t (1 - e_n^\top V_{s-}) J_n(s) dM_s^X = \\
&= \int_0^t J_n(s) \Lambda^\top(s) (I - \operatorname{diag} J_n(s)) X_s ds + \int_0^t (1 - e_n^\top V_{s-}) J_n(s) dM_s^X = \\
&= \int_0^t \mathbf{1} \Gamma_n(s) X_s ds + \int_0^t (1 - e_n^\top V_{s-}) J_n(s) dM_s^X. \quad \text{(A1)}
\end{aligned}
$$

Assertion 1 of Lemma is proved.

The definition of the processes $C_t^n$ ($n = \overline{1,N}$) guarantees their strong orthogonality, i.e., $\mathcal{P}\left\{ \Delta C_t^i \Delta C_t^j = 0 \right\} \equiv 0$ for any $i \neq j$ and $t \geqslant 0$, so $[C^i, C^j]_t \equiv 0$.

Let us use (5), (19) and properties of $X$ and $J_n$ to derive the quadratic characteristics of $C^n$:

$$
\begin{aligned}
\langle C^n, C^n \rangle_t &= \int_0^t (1 - J_n(s) X_{s-})^2 J_n(s) d\langle X, X \rangle_s J_n^\top(s) = \\
&= \int_0^t (1 - J_n(s) X_{s-}) J_n(s) \left( \operatorname{diag}(\Lambda^\top(s) X_{s-} - \Lambda^\top(s) \operatorname{diag} X_{s-} - \operatorname{diag}(X_{s-}) \Lambda(s) \right) J_n^\top(s) ds = \\
&= \int_0^t (1 - J_n(s) X_{s-}) J_n(s) \operatorname{diag}(J_n(s)) \Lambda^\top(s) X_{s-} ds = \int_0^t J_n(s) \Lambda^\top(s) (I - \operatorname{diag} J_n(s)) X_s ds = \\
&= \int_0^t \mathbf{1} \Gamma_n(s) X_s ds. \quad \text{(A2)}
\end{aligned}
$$

Assertion 2 of Lemma is proved.

If $s$ and $t$ are two arbitrary moments, such that $s \leqslant t$, then

$$
\begin{aligned}
\mathbf{E}\left\{ v_t^n - v_s^n | \overline{\mathcal{Y}}_s \right\} &= \mathbf{E}\left\{ \int_s^t J_n(u) \Lambda^\top(u) (I - \operatorname{diag} J_n(u)) \mathbf{E}\left\{ (X_u - \widehat{X}_u) | \overline{\mathcal{Y}}_u \right\} du | \overline{\mathcal{Y}}_s \right\} + \\
&\quad + \mathbf{E}\left\{ \mathbf{E}\left\{ \int_s^t (1 - J_n(s) X_{s-}) J_n(u) dM_u^X | \mathcal{F}_s \right\} | \overline{\mathcal{Y}}_s \right\} = 0,
\end{aligned}
$$

i.e., $v_t^n$ is a $\overline{\mathcal{Y}}_t$-adapted martingale. Note, that $v_t^n$ is purely discontinuous with unit jumps, hence

$$
\begin{aligned}
[v^n, v^n]_t &= \sum_{\tau \leqslant t} (\Delta v_\tau^n)^2 = [C^n, C^n]_t = \sum_{\tau \leqslant t} (\Delta C_\tau^n)^2 = C_t^n = \\
&= \int_0^t J_n(s) \Lambda^\top(s) (I - \operatorname{diag} J_n(s)) X_s ds + \int_0^t (1 - J_n(s) X_{s-}) J_n(s) dM_s^X = \int_0^t \mathbf{1} \Gamma_n(s) \widehat{X}_s ds + \mu_t^0,
\end{aligned}
$$

where $\mu_t^0$ is some $\overline{\mathcal{Y}}_t$-adapted martingale. From the uniqueness of the special martingale representation $[v^n, v^n]_t$ it follows that $\langle v^n, v^n \rangle_t = \int_0^t \mathbf{1} \Gamma_n(s) \widehat{X}_s ds$. Lemma 2 is proved. $\square$

## Appendix B. Proof of Theorem 1

We use the same approach as in ([6], Part III, Sect. 8.7) to derive the MJP filtering equations. The idea exploits the uniqueness of the representation for a special semimartingale along with the integral representation of a martingale [23].

From the Bayes rule it follows that $\widehat{X}_0 = \mathbf{E}\{X_0|D_0\} = \left(D_0^\top J(0)\pi\right)^+ \mathrm{diag}(D_0)J(0)\pi$. Let $\varkappa_{n-1}$ be a random instant of the $n-1$-th discrete observation $\Delta D_{\varkappa_{n-1}}$. We investigate evolution of $X_t$ over the interval $[\varkappa_{n-1}, \varkappa_n)$:

$$X_t = X_{\varkappa_{n-1}} + \int_{\varkappa_{n-1}}^t \Lambda^\top(s)X_s ds + M_t^X - M_{\varkappa_{n-1}}^X, \quad t \in [\varkappa_{n-1}, \varkappa_n).$$

Conditioning the left and right parts of the latter equality over $\overline{\mathcal{Y}}_t$, one can show that

$$\widehat{X}_t = \widehat{X}_{\varkappa_{n-1}} + \int_{\varkappa_{n-1}}^t \Lambda^\top(s)\widehat{X}_s ds + \mu_t^1,$$

where $\{\mu_t^1\}_{t\in[\varkappa_{n-1},\varkappa_n)}$ is an $\overline{\mathcal{Y}}_t$adapted martingale. For any $t \in [\varkappa_{n-1}, \varkappa_n)$ the equality $\overline{\mathcal{Y}}_t = \overline{\mathcal{Y}}_{\varkappa_{n-1}} \vee \sigma\{U_s, \ s \in (\varkappa_{n-1}, t]\} \vee \sigma\{C_s^j, \ s \in (\varkappa_{n-1}, t], \ j = \overline{1,N}\}$ holds. The process $\{\omega_t\}$ (24) is a $\overline{\mathcal{Y}}_t$ -adapted standard Wiener process [10].

The process $U_t$ is a $\overline{\mathcal{Y}}_t$-adapted semimartingale with $\mathcal{F}^X$-conditionally-independent increments, meanwhile $\{C_t^j\}_{j=\overline{1,N}}$ are $\overline{\mathcal{Y}}_t$-adapted point processes. Hence, the martingale $\mu_t^1$ admits an integral representation ([23], Chap. 4, §8, Problem 1), i.e.,

$$\widehat{X}_t = \widehat{X}_{\varkappa_{n-1}} + \int_{\varkappa_{n-1}}^t \Lambda^\top(s)\widehat{X}_s ds + \int_{\varkappa_{n-1}}^t \alpha_s d\omega_s + \int_{\varkappa_{n-1}}^t \sum_{j=1}^N \beta_s^j dv_s^j, \tag{A3}$$

where $\alpha_t$ and $\{\beta_t^j\}_{j=\overline{1,N}}$ are $\overline{\mathcal{Y}}_t$-predictable processes of appropriate dimensionality, which should be determined.

Due to the generalized Itô rule

$$X_t U_t^\top = X_{\varkappa_{n-1}} U_{\varkappa_{n-1}}^\top + \int_{\varkappa_{n-1}}^t \left(\Lambda^\top(s)X_s U_s^\top + \mathrm{diag}(X_s)\overline{f}^\top(s)\right) ds + \mu_t^2,$$

where $\mu_t^2$ is an $\mathcal{F}_t$-adapted matringale. Conditioning both sides of the latter equality over $\overline{\mathcal{Y}}_t$, we can show that

$$\widehat{X}_t U_t^\top = \widehat{X}_{\varkappa_{n-1}} U_{\varkappa_{n-1}}^\top + \int_{\varkappa_{n-1}}^t \left(\Lambda^\top(s)\widehat{X}_s U_s^\top + \mathrm{diag}(\widehat{X}_s)\overline{f}^\top(s)\right) ds + \mu_t^3, \tag{A4}$$

where $\mu_t^3$ is a $\overline{\mathcal{Y}}_t$-adapted martingale. On the other hand, using the Itô rule, representation (A3) and the fact that $\omega_t$ is the Wiener process, we can obtain

$$\widehat{X}_t U_t^\top = \widehat{X}_{\varkappa_{n-1}} U_{\varkappa_{n-1}}^\top + \int_{\varkappa_{n-1}}^t \left(\Lambda^\top(s)\widehat{X}_s U_s^\top + \widehat{X}_s \widehat{X}_s^\top \overline{f}^\top(s) + \alpha_s\right) ds + \mu_t^4, \tag{A5}$$

where $\mu_t^4$ is a $\overline{\mathcal{Y}}_t$-adapted martingale. One can see that (A4) and (A5) are two representations of the same special semimartingale $\widehat{X}_t U_t^\top$, hence due to the representation uniqueness the $\overline{\mathcal{Y}}_t$-predictable process $\alpha_t$ should satisfy the equality

$$\int_{\varkappa_{n-1}}^t \mathrm{diag}(\widehat{X}_s)\overline{f}^\top(s)ds = \int_{\varkappa_{n-1}}^t \left(\widehat{X}_s \widehat{X}_s^\top \overline{f}^\top(s) + \alpha_s\right) ds,$$

and $\alpha_t$ may be chosen in the form

$$\alpha_t = \left(\mathrm{diag}\,\widehat{X}_{t-} - \widehat{X}_{t-}\widehat{X}_{t-}^\top\right)\overline{f}^\top(t). \tag{A6}$$

Due to the generalized Itô rule, formulae (5), (18) and the properties of $X$ and $J_j$ we can obtain, that

$$X_t C_t^j = X_{\varkappa_{n-1}} C_{\varkappa_{n-1}}^j + \int_{\varkappa_{n-1}}^t \left( \Lambda^\top(s) X_s C_s^j + \Gamma_j(s) X_s \right) ds + \mu_t^5,$$

where $\mu_t^5$ is an $\mathcal{F}_t$-adapted martingale. Conditioning both sides of this equality over $\overline{\mathcal{Y}}_t$, we get

$$\widehat{X}_t C_t^j = \widehat{X}_{\varkappa_{n-1}} C_{\varkappa_{n-1}}^j + \int_{\varkappa_{n-1}}^t \left( \Lambda^\top(s) \widehat{X}_s C_s^j + \Gamma_j(s) \widehat{X}_s \right) ds + \mu_t^6, \tag{A7}$$

where $\mu_t^6$ is a $\overline{\mathcal{Y}}_t$-adapted martingale. On the other hand, using the Itô rule, representation (A3) and quadratic characteristic (21) we deduce, that

$$\widehat{X}_t C_t^j = \widehat{X}_{\varkappa_{n-1}} C_{\varkappa_{n-1}}^j + \int_{\varkappa_{n-1}}^t \left( \Lambda^\top(s) \widehat{X}_s C_s^j + \widehat{X}_s \mathbf{1} \Gamma_j(s) \widehat{X}_s + \beta_s^j \mathbf{1} \Gamma_j(s) \widehat{X}_s \right) ds + \mu_t^7, \tag{A8}$$

where $\mu_t^7$ is a $\overline{\mathcal{Y}}_t$-adapted martingale. Since the representations (A7) and (A8) correspond to the same special semimartingale $\widehat{X}_t C_t^j$ we conclude that the process $\beta_s^j$ should satisfy the equality

$$\int_{\varkappa_{n-1}}^t \Gamma_j(s) \widehat{X}_s ds = \int_{\varkappa_{n-1}}^t \left[ \widehat{X}_s \mathbf{1} \Gamma_j(s) \widehat{X}_s + \beta_s^j \mathbf{1} \Gamma_j(s) \widehat{X}_s \right] ds.$$

Acting as with the coefficient $\alpha_t$, we choose the predictable processes $\beta_t^j$ in the form

$$\beta_t^j = \left( \Gamma_j(t) - \mathbf{1} \Gamma_j(t) \widehat{X}_{t-} I \right) \widehat{X}_{t-} \left( \mathbf{1} \Gamma_j(t) \widehat{X}_{t-} \right)^+, \quad j = \overline{1, N}. \tag{A9}$$

So, on the interval $[\varkappa_{n-1}, \varkappa_n)$ the optimal filtering estimate $\widehat{X}_t$ is described by the SDS

$$\widehat{X}_t = \widehat{X}_{\varkappa_{n-1}} + \int_{\varkappa_{n-1}}^t \Lambda^\top(s) \widehat{X}_{s-} ds + \int_{\varkappa_{n-1}}^t (\operatorname{diag} \widehat{X}_{s-} - \widehat{X}_{s-} \widehat{X}_{s-}^\top) \overline{f}^\top(s) d\omega_s +$$
$$+ \sum_{j=1}^N \int_{\varkappa_{n-1}}^t \left( \Gamma_j(s) - \mathbf{1} \Gamma_j(s) \widehat{X}_{s-} I \right) \widehat{X}_{s-} \left( \mathbf{1} \Gamma_j(s) \widehat{X}_{s-} \right)^+ d\nu_s^j. \tag{A10}$$

Since $\mathcal{P} \{ \Delta X_{\varkappa_n} = 0 \} = 1$, equation (A10) presumes $\mathcal{P}$-a.s. fulfilment of the equality

$$\mathbf{E} \left\{ X_{\varkappa_n} | \overline{\mathcal{Y}}_{\varkappa_{n-1}} \vee \sigma\{U_s, \ s \in (\varkappa_{n-1}, \varkappa_n]\} \vee \sigma\{C_s^j, \ s \in (\varkappa_{n-1}, \varkappa_n], \ j = \overline{1, N}\} \right\} =$$
$$= \widehat{X}_{\varkappa_{n-1}} + \int_{\varkappa_{n-1}}^{\varkappa_n} \Lambda^\top(s) \widehat{X}_{s-} ds + \int_{\varkappa_{n-1}}^{\varkappa_n} (\operatorname{diag} \widehat{X}_{s-} - \widehat{X}_{s-} \widehat{X}_{s-}^\top) \overline{f}^\top(s) d\omega_s +$$
$$+ \sum_{j=1}^N \int_{\varkappa_{n-1}}^{\varkappa_n} \left( \Gamma_j(s) - \mathbf{1} \Gamma_j(s) \widehat{X}_{s-} I \right) \widehat{X}_{s-} \left( \mathbf{1} \Gamma_j(s) \widehat{X}_{s-} \right)^+ d\nu_s^j = \widehat{X}_{\tau_n-}.$$

Finally,

$$\overline{\mathcal{Y}}_{\varkappa_n} = \overline{\mathcal{Y}}_{\varkappa_{n-1}} \vee \sigma\{U_s, \ s \in (\varkappa_{n-1}, \varkappa_n]\} \vee \sigma\{C_s^j, \ s \in (\varkappa_{n-1}, \varkappa_n], \ j = \overline{1, N}\} \vee \sigma\{\Delta D_{\varkappa_n}\},$$

so, by the Bayes rule we get that

$$\widehat{X}_{\tau_n} = \left( \Delta D_{\tau_n}^\top \Delta J(\tau_n) \widehat{X}_{\tau_n-} \right)^+ \operatorname{diag}(\Delta D_{\tau_n}) \Delta J(\tau_n) \widehat{X}_{\tau_n-}. \tag{A11}$$

Equation (23) can be obtained as "gluing" of local equations (A10), which describe the evolution of $\widehat{X}_t$ on the intervals $[\varkappa_{n-1}, \varkappa_n)$, and formula (A11), which describes the estimate correction given the observations available at the moments $\varkappa_n$.

Uniqueness of the strong solution within the class of nonnegative piecewise-continuous $\mathcal{Y}_{t+}$-adapted processes with discontinuity set lying in $\mathcal{V}$ can be proved in complete analogy with ([31] Chap. 9, Theorem 9.2). Theorem 1 is proved. $\square$

## Appendix C. Proof of Corollary 1

The conditions of Corollary guarantee, that the elements of $K(t)$ (4) satisfy the equality $K_{nm}(t) = \delta_{nm}$ almost everywhere, hence $J(t) \equiv I$. This means that in (23) $D_0 = X_0$, $\mathcal{P}$-a.s., i.e., $\widehat{X}_0 = X_0$. Further, from the properties of transition intensity matrix $\Lambda(\cdot)$ and the identity $J_n(t) \equiv e_n^\top$ it follows that $\Gamma_n(t) = \operatorname{diag}(e_n)\overline{\Lambda}^\top(t)$, where $\overline{\Lambda}(t) \triangleq \Lambda(t) - \lambda(t)$, $\lambda(t) \triangleq \operatorname{diag}(\Lambda_{11}(t), \ldots, \Lambda_{NN})$. In this case

$$C_t = \int_0^t \overline{\Lambda}^\top(s) X_s ds + \int_0^t (I - \operatorname{diag} X_{s-}) dM_s^X,$$

and the $n$-th component counts the jumps of $X_t$ into the state $e_n$, occurred on the interval $(0, t]$. This means $X_t$ is the unique solution to the "purely discontinuous" equation

$$X_t = D_0 + \int_0^t (I - X_{s-}\mathbf{1}) dC_s, \tag{A12}$$

i.e., the state $X_t$ is measurable with respect to $\sigma\{D_0, C_s, 0 \leqslant s \leqslant t\}$, so $\widehat{X}_t = X_t$ $\mathcal{P}$-a.s.

Further, we substitute $X_t$ into (23) and verify its validity. To do this we simplify the RHS of the equality using the explicit form of $J_n(t)$, $\Gamma_n(t)$ and $C_t$, along with the identities $\operatorname{diag} X_t - X_t X_t^\top \equiv 0$ and $\Delta J(t) \equiv 0$:

$$X_t = D_0 + \int_0^t \Lambda^\top(s) X_s ds +$$

$$+ \sum_{n=1}^N \int_0^t \left[ \operatorname{diag}(e_n)\overline{\Lambda}^\top(s) - e_n^\top \overline{\Lambda}^\top(s) X_{s-} I \right] X_{s-} \left( e_n^\top \overline{\Lambda}^\top(s) X_{s-} \right)^+ \left[ dC_s^n - e_n^\top \overline{\Lambda}^\top(s) X_{s-} ds \right] =$$

$$= D_0 + \sum_{n=1}^N \int_0^t \left[ \operatorname{diag}(e_n)\overline{\Lambda}^\top(s) - e_n^\top \overline{\Lambda}^\top(s) X_{s-} I \right] X_{s-} \left( e_n^\top \overline{\Lambda}^\top(s) X_{s-} \right)^+ dC_s^n.$$

The properties of counting processes also provides the following implication: if for some $\mathfrak{T} \subseteq [0, T]$ the equality $\int_{\mathfrak{T}} e_n^\top \overline{\Lambda}^\top(s) X_s ds = 0$ holds, then $\int_{\mathfrak{T}} dC_s^n = 0$. Hence, the latter transformation can be continued:

$$X_t = D_0 + \sum_{n=1}^N \int_0^t [e_n - X_{s-}] e_n^\top dC_s = D_0 + \int_0^t (I - X_{s-}\mathbf{1}) dC_s,$$

which leads to (A12). So, we have verified that under conditions of Corollary 1 the state $X_t$ is a solution to the filtering equation (23). Corollary 1 is proved. $\square$

## Appendix D. Proof of Lemma 4

Using notations $\Xi_r \triangleq \xi_1 \xi_2 \ldots \xi_r$ and $\Theta_r \triangleq \theta_1 \theta_2 \ldots \theta_r$ we can rewrite the estimates $\widehat{X}_r$ and $\overline{X}_r(s)$ in the explicit form

$$\widehat{X}_r = \left( \mathbf{1} \left( \Xi_r + \Theta_r \right)^\top \pi \right)^{-1} \left( \Xi_r + \Theta_r \right)^\top \pi, \qquad \overline{X}_r(s) = \left( \mathbf{1}\Xi_r^\top \pi \right)^{-1} \Xi_r^\top \pi.$$

To simplify inferences we will omit the index $r$ in $\Xi_r$ and $\Theta_r$. The following relations are valid

$$\mathbf{E}\left\{\left\|\widehat{\mathsf{X}}_r - \overline{\mathsf{X}}_r(s)\right\|_1\right\} = \mathbf{E}\left\{\left\|\tfrac{1}{\mathbf{1}(\Xi+\Theta)^\top \pi}(\Xi+\Theta)^\top \pi - \tfrac{1}{\mathbf{1}\Xi^\top \pi}\Xi^\top \pi\right\|_1\right\} =$$

$$= \mathbf{E}\left\{\tfrac{1}{\mathbf{1}(\Xi+\Theta)^\top \pi \mathbf{1}\Xi^\top \pi}\left\|\mathbf{1}\Xi^\top \pi\Theta^\top \pi - \mathbf{1}\Theta^\top \pi\Xi^\top \pi\right\|_1\right\} \leqslant$$

$$\leqslant \mathbf{E}\left\{\tfrac{1}{\mathbf{1}(\Xi+\Theta)^\top \pi \mathbf{1}\Xi^\top \pi}\left(\mathbf{1}\Xi^\top \pi\|\Theta^\top \pi\|_1 + \mathbf{1}\Theta^\top \pi\|\Xi^\top \pi\|_1\right)\right\} = 2\mathbf{E}\left\{\tfrac{1}{\mathbf{1}(\Xi+\Theta)^\top \pi}\mathbf{1}\Theta^\top \pi\right\}. \quad (A13)$$

Let us consider an auxiliary estimate $\check{\mathsf{X}}_r \triangleq \mathbf{E}\left\{X_{t_r}\mathbf{I}_{A_r^s}(\omega)|\mathfrak{Y}_r\right\}$. From the Bayes rule it follows that $\check{\mathsf{X}}_r = \tfrac{1}{\mathbf{1}(\Xi+\Theta)^\top \pi}\Xi^\top \pi$ and

$$\widehat{\mathsf{X}}_r - \check{\mathsf{X}}_r = \mathbf{E}\left\{X_{t_r}\mathbf{I}_{\overline{A}_r^s}(\omega)|\mathfrak{Y}_r\right\} = \tfrac{1}{\mathbf{1}(\Xi+\Theta)^\top \pi}\Theta^\top \pi. \quad (A14)$$

From (A13) and (A14) we deduce, that for $r = 1$ and $\forall\, \pi \in \Pi$

$$\mathbf{E}\left\{\|\widehat{\mathsf{X}}_1 - \overline{\mathsf{X}}_1(s)\|_1\right\} \leqslant 2\mathbf{E}\left\{\|\mathbf{E}\left\{X_{t_1}\mathbf{I}_{\overline{a}_1^s}(\omega)|\mathfrak{Y}_1\right\}\|_1\right\} =$$

$$= 2\mathbf{E}\left\{\sum_{n=1}^{N}\mathbf{E}\left\{X_{t_1}^n\mathbf{I}_{\overline{a}_1^s}(\omega)|\mathfrak{Y}_1\right\}\right\} = 2\mathbf{E}\left\{\mathbf{E}\left\{\mathbf{I}_{\overline{a}_1^s}(\omega)|\mathfrak{Y}_1\right\}\right\} = 2\mathcal{P}\left\{\overline{a}_1^s\right\}. \quad (A15)$$

The counting process $N_t^X$ has the quadratic characteristic $\langle N^X, N^X\rangle_t = -\int_0^t \sum_{n=1}^{N}\lambda_{nn}X_s^n ds$, hence the probability $\mathcal{P}\left\{\overline{a}_1^s\right\}$ can be bounded from above as

$$\mathcal{P}\left\{\overline{a}_1^s\right\} \leqslant e^{-\overline{\lambda}h}\sum_{k=s+1}^{\infty}\tfrac{(\overline{\lambda}h)^k}{k!} = C_1\tfrac{(\overline{\lambda}h)^{s+1}}{(s+1)!}. \quad (A16)$$

Formulae (A15) and (A16) lead to the fact, that $\sup_{\pi\in\Pi}\mathbf{E}\left\{\|\widehat{\mathsf{X}}_1 - \overline{\mathsf{X}}_1(s)\|_1\right\} \leqslant 2C_1\tfrac{(\overline{\lambda}h)^{s+1}}{(s+1)!}$.

Markovianity of the pair $(X_t, N_t^X)$ and inequality (A16) also allow to bound the probability $\mathcal{P}\left\{\overline{A}_r^s\right\}$ from above: $\mathcal{P}\left\{\overline{A}_r^s\right\} \leqslant 1 - \left(1 - C_1\tfrac{(\overline{\lambda}h)^{s+1}}{(s+1)!}\right)^r$, that leads to (34). Lemma 4 is proved. □

**Appendix E. Proof of Theorem 2**

We have $\widetilde{\mathsf{X}}_1 = (\mathbf{1}\psi_1^\top \pi)^{-1}\psi_1^\top \pi$, $\overline{\mathsf{X}}_1 = (\mathbf{1}\xi_1^\top \pi)^{-1}\xi_1^\top \pi$ and $\Delta_1 = \widetilde{\mathsf{X}}_1 - \overline{\mathsf{X}}_1(s)$. Using the matrix algebra it is easy to verify that $[\gamma^\top \mathbf{1} - \mathbf{1}\gamma^\top \pi I]\gamma^\top \pi \equiv 0$. Both the estimates are stable, hence $\|\widetilde{\mathsf{X}}_1\|_1 = \|\overline{\mathsf{X}}_1(s)\|_1 = 1$. The following relations are valid:

$$\|\Delta_1\|_1 = \tfrac{1}{\mathbf{1}\psi_1^\top \pi \mathbf{1}\xi_1^\top \pi}\|\mathbf{1}\xi_1^\top \pi\psi_1^\top \pi - \mathbf{1}\psi_1^\top \pi\xi_1^\top \pi\|_1 = \tfrac{1}{\mathbf{1}\psi_1^\top \pi \mathbf{1}\xi_1^\top \pi}\|\mathbf{1}\xi_1^\top \pi\gamma_1^\top \pi - \mathbf{1}\gamma_1^\top \pi\xi_1^\top \pi\|_1 =$$

$$= \tfrac{1}{\mathbf{1}\psi_1^\top \pi \mathbf{1}\xi_1^\top \pi}\|[\gamma_1^\top \pi\mathbf{1} - \mathbf{1}\gamma_1^\top \pi I]\xi_1^\top \pi\|_1 =$$

$$= \tfrac{1}{\mathbf{1}\psi_1^\top \pi \mathbf{1}\xi_1^\top \pi}\|[\gamma_1^\top \pi\mathbf{1} - \mathbf{1}\gamma_1^\top \pi I][\xi_1^\top \pi + \gamma_1^\top \pi]\|_1 = \tfrac{1}{\mathbf{1}\xi_1^\top \pi}\|[\gamma_1^\top \pi\mathbf{1} - \mathbf{1}\gamma_1^\top \pi I]\widetilde{\mathsf{X}}_1\|_1 \leqslant$$

$$\leqslant \tfrac{1}{\mathbf{1}\xi_1^\top \pi}\|[\gamma_1^\top \pi\mathbf{1} - \mathbf{1}\gamma_1^\top \pi I]\|_1\|\widetilde{\mathsf{X}}_1\|_1 \leqslant 2\tfrac{\mathbf{1}\overline{\gamma}_1^\top \pi}{\mathbf{1}\xi_1^\top \pi} = \sum_{i=1}^{N}\pi_i\tfrac{\sum_{j=1}^{N}\overline{\gamma}_1^{ij}}{\sum_{k,\ell=1}^{N}\xi_1^{k\ell}\pi_k}.$$

Using the last inequality, (41) and (A20), it can be shown that

$$\mathbf{E}\left\{\mathbf{I}_{a_1^s}(\omega)\|\Delta_1\|_1\right\} \leqslant 2\sum_{i=1}^{N}\pi_i\int_{\mathbb{R}^M}\sum_{i=1}^{N}\overline{\gamma}^{ij}(y)dy \leqslant 2\delta.$$

Since the latter inequality is valid for any $\pi \in \Pi$, we have an upper bound for the local distance characteristic:

$$\sup_{\pi \in \Pi} \mathbf{E} \left\{ \mathbf{I}_{a_1^s}(\omega) \| \widetilde{X}_1 - \overline{X}_1(s) \|_1 \right\} \leqslant 2\delta. \tag{A17}$$

Let us define the following products of the random matrices $\xi_r$ and $\psi_r$:

$$\Xi_{q,r} \triangleq \begin{cases} \xi_q \xi_{q+1} \dots \xi_r, & \text{if } q \leqslant r, \\ I & \text{otherwise,} \end{cases}$$

$$\Psi_{q,r} \triangleq \begin{cases} \psi_q \xi_{q+1} \dots \psi_r, & \text{if } q \leqslant r, \\ I & \text{otherwise,} \end{cases}$$

$$\Gamma_{q,r} \triangleq \Psi_{q,r} - \Xi_{q,r}.$$

To proceed the proof of Theorem 2 we need the following auxiliary

**Lemma A1.** *If $\phi_r \triangleq \phi_r(Y_1, \dots, Y_r)$ is a non-negative $\mathfrak{Y}_r$-measurable random value, and $\Phi_r \triangleq \frac{\phi_r}{\mathbf{1}\Xi_{1,r}^\top \pi}$, then*

$$\mathbf{E} \left\{ \mathbf{I}_{A_r^s}(\omega) \Phi_r \right\} = \int_{\mathbb{R}^M} \dots \int_{\mathbb{R}^M} \phi_r(y_1, \dots, y_r) dy_r \dots dy_1. \tag{A18}$$

**Proof of Lemma A1.** We consider a non-negative integrable function $\phi_1 = \phi_1(y) : \mathbb{R}^M \to \mathbb{R}_+$ and a $\mathfrak{Y}_1$-measurable random value

$$\Phi_1 \triangleq \frac{\phi_1(Y_1)}{\mathbf{1}\xi_1^\top(Y_1)\pi} = \frac{\phi_1(Y_1)}{\sum_{i,j=1}^N \sum_{m=0}^s \int_{\mathcal{D}} \mathcal{N}(Y_1, fu, \sum_{p=1}^N u^p G_p) \rho^{i,j,m}(du) \pi_i}. \tag{A19}$$

We find $\mathbf{E} \left\{ \mathbf{I}_{a_1^s}(\omega) \Phi_1 \right\}$:

$$\mathbf{E} \left\{ \mathbf{I}_{a_1^s}(\omega) \Phi_1 \right\} = \int_{\mathbb{R}^M} \int_{\mathcal{D}} \frac{\phi_1(y) \sum_{k,\ell=1}^N \sum_{n=0}^s \mathcal{N}(y, fv, \sum_{q=1}^N v^q G_q) \rho^{k,\ell,n}(dv) \pi_k}{\sum_{i,j=1}^N \sum_{m=0}^s \int_{\mathcal{D}} \mathcal{N}(y, fu, \sum_{p=1}^N u^p G_p) \rho^{i,j,m}(du) \pi_i} dy =$$

$$= \int_{\mathbb{R}^M} \phi_1(y) \frac{\sum_{k,\ell=1}^N \sum_{n=0}^s \int_{\mathcal{D}} \mathcal{N}(y, fv, \sum_{q=1}^N v^q G_q) \rho^{k,\ell,n}(dv) \pi_k}{\sum_{i,j=1}^N \sum_{m=0}^s \int_{\mathcal{D}} \mathcal{N}(y, fu, \sum_{p=1}^N u^p G_p) \rho^{i,j,m}(du) \pi_i} dy = \int_{\mathbb{R}^M} \phi_1(y) dy. \tag{A20}$$

Let us consider a non-negative integrable function $\phi_2 = \phi_1(y_1, y_2) : \mathbb{R}^{2M} \to \mathbb{R}_+$ and a $\mathfrak{Y}_2$-measurable random value

$$\Phi_2 \triangleq \frac{\phi_1(Y_1, Y_2)}{\mathbf{1}\Xi_{1,2}^\top(Y_1, Y_2)\pi} =$$

$$= \frac{\phi_2(Y_1, Y_2)}{\sum_{i,i_2,j=1}^N \sum_{m_1,m_2=0}^s \int_{\mathcal{D}} \int_{\mathcal{D}} \mathcal{N}(Y_1, fu_1, \sum_{p_1=1}^N u^{p_1} G_{p_1}) \mathcal{N}(Y_2, fu_2, \sum_{p_2=1}^N u^{p_2} G_{p_2}) \rho^{i,i_2,m_1}(du_1) \rho^{i_2,j,m_2}(du_2) \pi_i}.$$

We find $\mathbf{E}\left\{\mathbf{I}_{A_2^s}(\omega)\Phi_2\right\}$:

$$\mathbf{E}\left\{\mathbf{I}_{A_2^s}(\omega)\Phi_2\right\} = \int_{\mathbb{R}^M}\int_{\mathbb{R}^M}\phi_2(y_1,y_2)\times$$

$$\times \frac{\displaystyle\sum_{k,k_2,\ell=1}^N\sum_{n_1,n_2=0}^s\int_{\mathcal{D}}\int_{\mathcal{D}}\mathcal{N}(y_1,fv_1,\sum_{q_1=1}^N v^{q_1}G_{q_1})\mathcal{N}(y_2,fv_2,\sum_{q_2=1}^N v^{q_2}G_{q_2})\rho^{k,k_2,n_1}(dv_1)\rho^{k_2,\ell,n_2}(dv_2)\pi_k}{\displaystyle\sum_{i,i_2,j=1}^N\sum_{m_1,m_2=0}^s\int_{\mathcal{D}}\int_{\mathcal{D}}\mathcal{N}(y_1,fu_1,\sum_{p_1=1}^N u^{p_1}G_{p_1})\mathcal{N}(y_2,fu_2,\sum_{p_2=1}^N u^{p_2}G_{p_2})\rho^{i,i_2,m_1}(du_1)\rho^{i_2,j,m_2}(du_2)\pi_i}dy_2dy_1 =$$

$$= \int_{\mathbb{R}^M}\int_{\mathbb{R}^M}\phi_2(y_1,y_2)dy_2dy_1.$$

The correctness of the Lemma assertion in the general case of $\mathbf{E}\left\{\mathbf{I}_{A_r^s}(\omega)\Phi_r\right\}$ can be verified similarly. Lemma A1 is proved. $\square$

Let us define an upper estimate for the norm of $\Delta_r = \widetilde{\mathsf{X}}_r - \overline{\mathsf{X}}_r$. From the definitions of $\Xi$, $\Psi$ and $\Gamma$ it follows that

$$\Gamma_{1,r} \triangleq \Psi_{1,r} - \Xi_{1,r} = \sum_{t=1}^r \Psi_{1,t-1}\gamma_t\Psi_{t+1,r}. \tag{A21}$$

Making the same inferences as for $\Delta_1$, we can deduce that

$$\|\Delta_r\|_1 \leqslant \tfrac{1}{\mathbf{1}\Xi_{1,r}^\top\pi}\|[\Gamma_{1,r}^\top\pi\mathbf{1} - \mathbf{1}\Gamma_{1,r}^\top\pi I]\|_1 \leqslant 2\sum_{t=1}^r \tfrac{1}{\mathbf{1}\Xi_{1,r}^\top\pi}\mathbf{1}\Psi_{t+1,r}^\top\overline{\gamma}_t^\top\Psi_{1,t-1}^\top\pi. \tag{A22}$$

To estimate the contribution of each summand in (A22) we use (A18). To simplify derivation we consider the case $r = 3$, function $\phi(y_1,y_2,y_3) : \mathbb{R}^{3M} \to \mathbb{R}_+$

$$\phi(y_1,y_2,y_3) = \mathbf{1}\psi^\top(y_3)\overline{\gamma}^\top(y_2)\psi^\top(y_1)\pi$$

and the $\mathfrak{Y}_3$-measurable random value $\Phi \triangleq \frac{\phi(Y_1,Y_2,Y_3)}{\mathbf{1}\Xi_{1,3}^\top(Y_1,Y_2,Y_3)\pi}$. Let us estimate from above the mathematical expectation

$$\mathbf{E}\left\{\mathbf{I}_{A_3^s}(\omega)\Phi\right\} = \int_{\mathbb{R}^M}\int_{\mathbb{R}^M}\int_{\mathbb{R}^M}\sum_{i,j,k,m=1}^N \pi_i\psi^{ij}(y_1)\overline{\gamma}^{jk}(y_2)\psi^{km}(y_3)dy_3dy_2dy_1 =$$

$$= \sum_{i,j,k=1}^N \pi_i\sum_{\ell=1}^L \varrho_\ell^{ij}\int_{\mathbb{R}^M}\overline{\gamma}^{jk}(y_2)dy_2\sum_{m=1}^N\sum_{n=1}^L \varrho_n^{km} = Q\sum_{i,j=1}^N \pi_i\sum_{\ell=1}^L \varrho_\ell^{ij}\sum_{k=1}^N\int_{\mathbb{R}^M}\overline{\gamma}^{jk}(y_2)dy_2 \leqslant$$

$$\leqslant Q\delta\sum_{i=1}^N \pi_i\sum_{j=1}^N\sum_{\ell=1}^L \varrho_\ell^{ij} \leqslant Q^2\delta.$$

Acting in the same way, we can prove that for arbitrary $r \geqslant 2$ the inequality

$$\mathbf{E}\left\{\mathbf{I}_{A_r^s}(\omega)\frac{\mathbf{1}\Psi_{t+1,r}^\top\overline{\gamma}_t^\top\Psi_{1,t-1}^\top\pi}{\mathbf{1}\Xi_{1,r}^\top\pi}\right\} \leqslant Q^{r-1}\delta$$

is valid for all $r$ summands in the RHS of (A22). Finally $\mathbf{E}\left\{\mathbf{I}_{A_r^s}(\omega)\|\Delta_r\|_1\right\} \leqslant 2rQ^{r-1}\delta$, and the correctness of (42) follows from the fact that the latter inequality is valid for arbitrary $\pi \in \Pi$. Theorem 2 is proved. $\square$

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
