# Peer review of "Optimal Filtering of Markov Jump Processes Given Observations with State-Dependent Noises: Exact Solution and Stable Numerical Schemes"

_mathematics, doi:10.3390/math8040506_

Round 1

Reviewer 1 Report

Comments to the authors are described in the file.

Author Response

Response to Reviewer 1 Comments

Point 1: 
 In spite of the merits of the proposed algorithm, it is not enough to show availability and usefulness for readers with satisfaction Namely, it is important to demonstrate /verify performance (accuracy/stability) of the present algorithm by displaying some sound numerical examples. Otherwise, it is difficult to attracting reader’s attentions. I would like to recommend the paper to accept after minor revision on this point.

Response 1: We have added subsection 4.4 “Numerical Example”. We hope this simple illustration helps readers to appreciate the properties of both the presented theoretical solution to the optimal filtering problem and the algorithm of its numerical realization.

Point 2: Typos are observed in the text, please correct them. E.g.,

Page 3, Line 86, col pi_1, pi_2, pi_N right arrow col ( pi_1, pi_2,pi_N )

Page 17, line 300, there are typos.

Also, many typos can be seen in references.

Response 2: We have corrected all the typos specified above. We also have verified the text thoroughly and hope to have wiped out all the misprints.

We are very grateful to Reviewer 1 for the high evaluation of our research and useful suggestions. We've done our best to correct the paper according to the Reviewer's comments.

Reviewer 2 Report

The article is describes to the optimal state filtering of the finite-state Markov jump processes, given indirect continuous-time observations corrupted by Wiener noises. We propose an equivalent observation transform, which allows usage of the classical nonlinear filtering framework.

The work is original and I recommend it for publication.

Author Response

Response to Reviewer 2 Comments

We are very grateful to Reviewer 2 for the high evaluation of our research.

Reviewer 3 Report

The article is presented in a very professional manner. I very rarery get to review a manuscript of such high level. I only have one remark on the reference list. Very old publications are included. Are there no newer ones? In my opinion, the authors should add at least 10 publications that appeared in recent years to the reference list.

Author Response

Response to Reviewer 3 Comments

Point 1: 
 Very old publications are included. Are there no newer ones? In my opinion, the authors should add at least 10 publications that appeared in recent years to the reference list.

Response 1: We have included just a few recent papers in the new variant of the article (please, see [8,9,21,22]). Unfortunately, there is only a small part of the papers addressed to the optimal state filtering in the stochastic differential systems, given the observations with state-dependent (aka multiplicative) noise. The avalanche of the phrase «state-dependent/multiplicative noise» in the titles of such papers is rather illusory. Mostly, suchlike papers do not relate to the problem considered in our article due to the following reasons:

  • they are devoted to a different problem rather than investigated in our paper: usually this is a control problem,
  • they investigate either discrete-time stochastic observation systems or ones with discrete-time observations,
  • the multiplicative noise places in the system dynamics not in the observations.

Currently, the researchers pay not too much attention to the estimation problem in the settings, as in our paper. The reason is not in the low practical value of the problem but its known mathematical complexity. We have attempted to expose the latter fact in Introduction.

We are very grateful to Reviewer 3 for the high evaluation of our research and useful suggestions. We've done our best to correct the paper according to the Reviewer's comments.
